# DDNB—Doubly Decentralized Network Blockchain Architecture for Application Services [†]

**Youhwan Seol [1]**, **Jaehong Ahn [2]**, **Sehyun Park [3]**, **Mookeun Ji [3]**, **Heungseok Chae [4]**, **Jiheon Yi [4]** **and Jeongyeup Paek [1,\*]**

[1] Department of Computer Science and Engineering, Chung-Ang University, Seoul 06947, Korea; adaan12@cau.ac.kr

[2] School of European Languages and Cultures, Chung-Ang University, Seoul 06947, Korea; ahong94@cau.ac.kr (J.A.); wlanrms95@cau.ac.kr (M.J.)

[3] School of Computer Science and Engineering, Chung-Ang University, Seoul 06947, Korea; po2562@cau.ac.kr

[4] ABC Inc., Seoul 08505, Korea; seok@4intel.net (H.C.); baupa@4intel.net (J.Y.)

[\*] Correspondence: jpaek@cau.ac.kr

[†] An earlier preliminary version of this concept was presented at the International Conference on ICT Convergence (ICTC) 2019. This manuscript adds the implementation, evaluation, and case-study of the concept built into a real working system.

**Abstract:** Decentralization and immutability characteristics of blockchain technology has attracted numerous blockchain-based systems and applications to be proposed. However, technical shortcomings such as low transaction speed, complexity, scalability, and vulnerability to certain attacks have been identified, making it challenging to use the technology on general consumer applications and services. To address the problem, we propose a new application service platform architecture called DDNB (Doubly Decentralized Network Blockchain). DDNB divides the system into multiple layers in order to take advantage of permissioned blockchain for its processing speed and security, while allowing permissionless open use of the system to application developers. To allow any node to freely participate in application services, DDNB adopts a novel periodic *node self-verification* process and *query chaining* mechanism to authenticate newly joining nodes and validate transactions effectively and efficiently. The proposed architecture is evaluated in terms of its processing speed and security on a real proof-of-concept prototype system.

**Keywords:** blockchain; decentralization; hierarchical architecture; application platform

## 1. Introduction

Bitcoin is one of the most representative cryptocurrency, and has contributed to building a reliable and decentralized cryptocurrency environment on a P2P network without the need of central trusted authorities [1]. Blockchain is the underlying data structure technology of Bitcoin, which is a continuously growing chain of blocks each having a set of transactions that occur between participating peers. Each participant maintains a distributed ledger consisting of sequentially chained blocks, and these blocks are propagated and validated by full nodes with the proof-of-work consensus protocol (in Bitcoin) between validator nodes. When an untrusted third party participates in the service operation, blockchain suppresses any possible data manipulation through the consensus process between the peers.

Characteristics such as immutability, irreversibility, and decentralization has attracted increasing interest in blockchain for purposes other than cryptocurrency, and numerous blockchain-based systems and applications have been proposed [2–17]. However, the architecture of the early blockchain systems,

such as Bitcoin, also had several drawbacks which made it difficult to apply to applications demanding more sophisticated operations than just 'transfer-of-ownership'. That is because most core parts of the system, like data structure and operations, are designed specifically for the purpose of cryptocurrency to avoid various network attacks (e.g., 51% majority attack).

Despite the drawbacks, the potential of blockchain technology shown by Bitcoin has led to a number of studies to improve it as a distributed or decentralized platform applicable to diverse types of application. By applying the '*smart-contract*' concept which makes it possible to derive deterministic and immediate output under pre-defined conditions, the blockchain technology seemed effective for developing decentralized applications (*DApps*). However, a fundamental problem driven by the consensus protocol still remained; the network was far too slow to implement practical applications at a large scale. Although the Proof-of-Work consensus protocol is an excellent way to deter arbitrary attackers in an public environment with ruthless attack potential, it works as an obstacle upon the DApps, restricting it from providing general services to users.

There have been several studies for more decisive but generous consensus protocols working on a permissioned/private network [18–21], but these attempts still have the limits of scalability as the network governs the access of the nodes to demarcate from outside. That is, the scalability and performance are incompatible trade-off factors in running an application on blockchain based platforms. In addition to the limitations mentioned above, service providers encounter several difficulties when implementing actual applications running on blockchain based platforms. Service providers not only need to handle the business logic specific to their services, but also tasks of designing and maintaining blockchain systems. In Section 2, the problems that have been encountered to apply blockchain for DApps will be discussed in more detail.

This paper presents 'Doubly Decentralized Network Blockchain' (DDNB), a platform architecture that suppresses untrusted participants' malicious behavior by separating the service and blockchain layers, and allows the service providers to conveniently implement general business logic on DApps. In DDNB, blockchain layer and the service layer are architecturally separated so that they depend only on each other's interfaces. While using the 'chaining' and 'distributed' concepts of blockchain, it provides a structure for service developers to develop DApps without the need of deeply understanding the blockchain technology.

## 2. Problem & Motivation

Blockchain has received explosive interest following the success of Bitcoin presented in Satoshi Nakamoto's paper [1]. Then, as the first blockchain based platform which supports smart contract, Ethereum [22] has brought the possibility of immutability and decentralization to various fields. Furthermore in Hyperledger Fabric [23], the network consists of permissioned nodes working upon the 'Practical Byzantine Fault Tolerance' algorithm as its consensus protocol, providing faster and cheaper finality for the blockchain system. Many attempts to improve blockchain have allowed the technology to be used to ensure reliability of arbitrary data in a decentralized network, more than just being used for cryptocurrency. For example, blockchain technology has been used in industry fields such as Internet of Things [2–6,24], E-commerce [14–16], Health Care [7–10], and Digital Rights Management [11–13] to take advantage of its properties for developing a variety of services.

However, there are several challenges that need to be addressed before applying blockchain technology on industrial applications that are developed for server-client systems in general. First and the most critical is the low transaction processing speed. Bitcoin adopted PoW (Proof-of-Work) mechanism for its block consensus algorithm. This process requires huge amount of computing power from participating full nodes (The nodes that can verify all of the rules of Bitcoin) in blockchain network, and it limits block creation cycle to 10 min on average (for Bitcoin). Although PoW has the great advantage of substantial ability in suppressing arbitrary participants' malicious behavior, the amount of resources wasted in the race for creating blocks is not negligible.

Second is the issue of scalability [25]. Blockchain is literally a chain of blocks sequentially linked by the hash value of previous block which contains a set of transactions. To maintain a consistent chain of blocks, a consensus process between network participants is required. Hence, system performance degrades significantly as the total network size increases due to the increase in amount of data that need to be exchanged. If the block size is increased to contain more transactions per block, the amount of information in a block can expand but the total propagation delay including the block verification time required for synchronizing blocks between nodes increases [26,27]. The longer the time spent for network synchronization, the longer the blockchain states remain inconsistent among the participants. This can make the blockchain network vulnerable to attacks such as *double spending*. A remedy to solve these defects is to establish a private network with permissioned nodes. By eliminating the possibility of malicious user threats in advance, fastidious consensus protocols are no longer needed. Instead, it is able to use optimized consensus algorithm within trusted environment resulting in increase of throughput.

Third is related to the irreversibility of blockchain. Blocks recorded upon consensus in blockchain network cannot be modified (without a full fork) even if the parties involved in the transaction agreed to change. While the irreversibility guarantees transparency of blockchain, it works as a critical flaw against several applications that need to provide certain reversible services. For example, when a smart contract is automatically executed upon certain preset conditions, there is no way to reverse the changes even if the outcome was an unexpected result due to a (malicious) bug in the smart contract code.

Last one is the complexity of blockchain system. Due to the complex logic of blockchain and issues mentioned above, arbitrary developers who want to launch application services with smart contract cannot concentrate entirely on their own business logic, and it is required for them to have advanced knowledge on the blockchain technology. Thus, it is still challenging to adopt blockchain technology to applications. But still, numerous attempts are being made to take advantage of its benefits. In this respect, DDNB would be a troubleshooter which allows to utilize the existing strong points of blockchain and ensure sufficient service reliability at the same time. Table 1 shows a high-level comparison of four different architectures that can be used for application development; the traditional server-client architecture, permissioned and permissionless blockchain, and our DDNB. Each approach has their own pros and cons, but our DDNB has its strong point in conducting business logic as fully-supported as in traditional development while it takes all advantages of private blockchain.

**Table 1.** Comparison of four different architectures' properties.

| Architectures<br>Properties | Server-Client | Permissionless B.C. * | Permissioned B.C. | DDNB |
|---|---|---|---|---|
| Processing Speed | Very-high | Very-low | High | High |
| Scalability | Very-easy | Very-limited | Very-limited | Less-limited |
| Business Logic | Fully supported | Very-limited | Limited | Fully supported |
| Data Integrity | Not sure, vulnerable | Very-robust | Robust | Robust |

* B.C. = Blockchain.

## 3. Related Work

This section explores several existing studies related to various attempts for adapting blockchain to applications. In particular, we surveyed three directions; (1) stability or performance of the network nodes maintaining blockchains, (2) new blockchain platforms for applications, and (3) applications on the basis of existing blockchain platforms such as Ethereum and Hyperledger.

### 3.1. Blockchain Network

At a high-level, a blockchain network is (mostly) a peer-to-peer (P2P) overlay network that changes dynamically and exchanges messages continuously. Nodes can join and leave at any time, and the protocols

work in a completely distributed manner. To understand this network, Decker et al. experimentally observed that increasing network delays with PoW leads to increased forks, which means an inconsistent state of the blockchain [26]. In 2014, a list of 872,648 different IP addresses known to run Bitcoin node are revealed by Donet et al. They used a client called 'BTCdoNET', a modified version of Bitcoin P2P Network Sniffer, and presented information on the geographic distribution, network stability, and information propagation latency [28]. Most recently, Park et al. collected information of participants of Bitcoin network for 39 days, and analyzed geographical distribution, protocol/client version, and the type of nodes (full-node vs. lightweight-node) [29].

### 3.2. Blockchain Platform

Among several proposals, HyperLedger presented by the Linux Foundation incubates and promotes a range of business blockchain technologies. Hyperledger includes Fabric; an enterprise grade permissioned distributed ledger framework, Besu; an open source Ethereum client written in Java, Sawtooth; a modular platform for building, deploying, and running distributed ledgers and so on. Changes to the consensus protocol are also significant developments. Delegated Proof-of-Stake literally delegates to the 'major node' determined by the voting result of all nodes in the network. The low number of major nodes reduces consensus time and costs [30–33].

To take advantage of P2P distributed nature of blockchain, several attempts to apply it on IoT systems [34] have emerged. In particular, as the early blockchain systems require a lot of computing resources and huge storage, most IoT related researches have focused on lightening the system or adopting additional module to fit in the hardware limits of IoT devices. Dorri et al. suggests an a lightweight blockchain based architecture for IoT which maintains most of legacy blockchain's security and privacy benefits while virtually eliminating the overheads [35]. This architecture also implemented a hierarchical structure; smart home, overlay network, and cloud storage; with *Cluster Head*, a cluster of constituent nodes. In a similar way, Novo et al. proposed a decentralized access management system where access control information is stored and distributed using blockchain technology [4]. They separated wireless sensor networks (WSN) from blockchain network and implemented a *Management Hub* which is an interface between the two. More recently, Lei et al. proposed Groupchain [36], a public blockchain of a two-chain structure designed for fog computing of IoT services by taking advantage of the security of blockchain while enhancing scalability. It employs the leader group to collectively commit blocks for higher transaction efficiency and introduces incentive mechanism to supervise behaviors of members in the leader group.

Most of these researches focused on addressing the resource-constraint challenge of IoT devices. However, to decrease blockchain network overhead, they compromised the distributed aspect with an aspect of server-client structural. It is a significant difference to DDNB which has no single-point standing for other things (e.g., devices, nodes). Although DDNB is not designed specifically to be well-suited for IoT devices, it is still able to adopt those attempts while assuring distributed Service Nodes. For example, it can be imagined that the additional nodes for certain IoT networks such as *Cluster Head* or *Management Hub* run on our service layer of DDNB. DDNB is possible to scale up, not only scale out.

### 3.3. Distributed Application

There has been several attempts to build distributed applications on top of blockchain. For example, Herbert et al. presented a decentralized P2P software validation scheme using blockchain where a user purchasing the license of a software sends cryptocurrency to the vendor on Bitcoin or Bespoke model [37]. Schaubs et al. suggested a blockchain-based trustless reputation system where every user of the system evaluates each other after a transaction, and all the evaluation data are safely stored in the blockchain to prevent manipulation [38]. Xia et al. proposed an access control platform based on blockchain which utilizes blockchain's immutability to record and keep track of every access to patients' medical information data [10]. For this purpose, the structure for blocks and transactions

were re-designed to store medical information. More recently, Zhaofeng et al. proposed BlockTDM [39], a blockchain-based trusted data management scheme for edge computing. BlockTDM is a configurable blockchain architecture that includes mutual authentication protocol, flexible consensus, smart contract, block/transaction data management, and blockchain nodes management to provide trust and security in edge computing environment for the large amount of data gathered from edge terminals or Internet of Things (IoT) devices.

Aforementioned are just a few examples, and there are/were many more attempts. However, most blockchain-based application developers face the common problems: they need to understand how blockchain system works or build a blockchain platform which is specially designed to provide specific services. With our 'DDNB', they can focus on implementing the service logic they want to provide to clients while utilizing all of the blockchain's properties.

## 4. Design of DDNB

DDNB is composed of two separated decentralized networks, one is a blockchain network that forms the distributed ledger, and the other is a service network that performs application business logic. This architecture allows service providers to treat service network as a server-side endpoint and alleviate concerns about the blockchain network. Compared to prior distributed applications that implement their services directly in blockchain network using smart contract, DDNB reduces the fundamental difficulties that need to be considered. To do so, however, it is required for the nodes right above blockchain layer to prove their reliability and trustworthiness. To accomplish this in DDNB, service layer nodes self-verify each other by going through a mutual verification process. Overlaying blockchain layer with verified nodes allows the blockchain nodes to remain as permissioned-only without any additional verification.

### 4.1. Node Constitution

DDNB consists of 3 layers: Terminal layer, Service layer and Blockchain layer, as illustrated in Figure 1. Each layer consists of different type of nodes respectively.

**Terminal Node (TN)** takes on a client-side user application role. It passes requests from users to Service Nodes (SN) to provide a specific service. Depending on how service providers, which consist of TNs and SNs, are designed, a number of logics can be implemented diversely.

**Service Node (SN)** provides and processes actual services. Therefore, SN is where actual service is designed and implemented. SN can act like a server in traditional server-client architecture, and so is the scope of logics which can be implemented in SN is almost unlimited. Thus, various types of services can be provided, more than any smart contracts can do. SN performs a pre-defined function according to the request of TN. It is responsible for delivering the results of the execution to Blockchain Node (BN), and returning them to TN. In this process, no individual data are stored in SN. For this reason, unlike typical web applications, session and sensitive information are managed in TN, not in SN.

**Blockchain Node (BN)** is the actual database for storing service related data. DDNB is designed to use permissioned blockchain so that BN can provide high transactions per second (TPS) and security while maintaining characteristics of blockchain. Also, by placing BNs behind VPN and firewall, it is possible to access control SNs to only those that are authorized. Therefore, service providers do not need to build their own nodes to participate in the blockchain. Instead, they only need to consider the consortium of the service networks. There can be multiple service networks consisting of individual domains to provide different services on a single blockchain network.

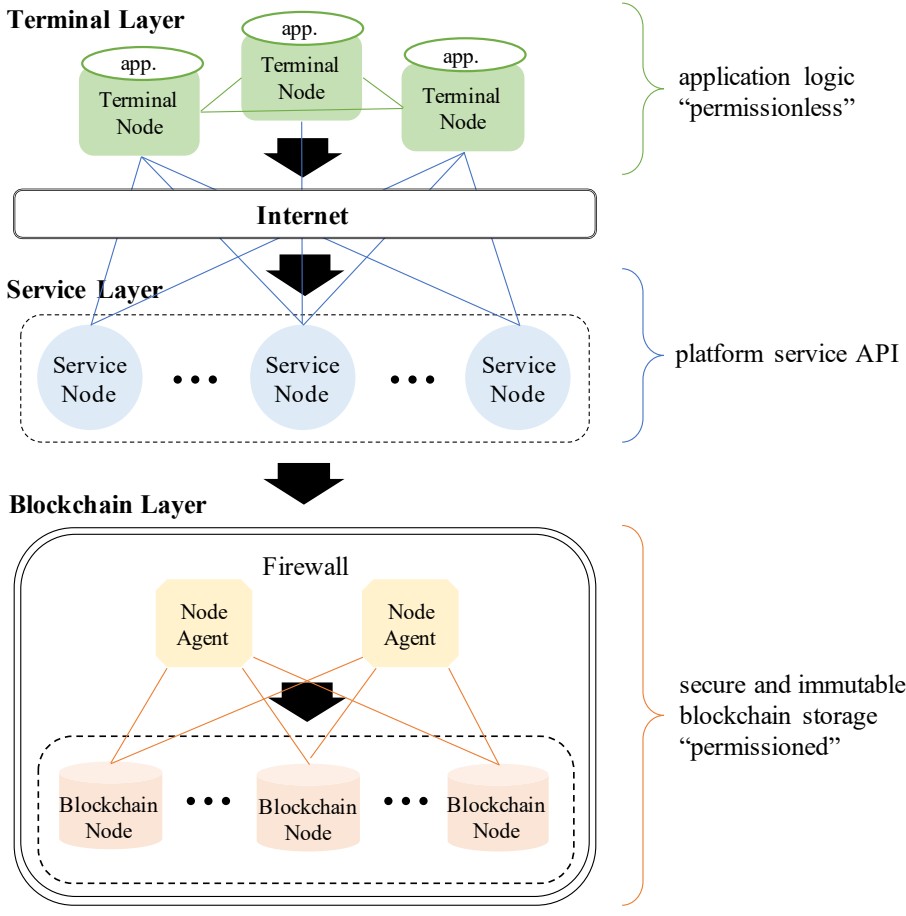

**Figure 1.** Three-layered Hierarchical Architecture of Doubly Decentralized Network Blockchain (DDNB).

*4.2. DDCP (Dynamic Decentralized Certification Protocol)*

SN provides pre-defined services to TN, and TN generates specific transactions to use those services. Before those transactions can access data in BN, DDNB verifies the integrity of the transaction through DDCP (Dynamic Decentralized Certification Protocol). In DDCP, TN segments a transaction logic into several query stages and sends a first query to a randomly selected SN. The selected SN executes one of the divided processes sequentially as shown in Figure 2. During this process, BN generates a short hashed string, *nonce*. DDCP uses this *nonce* to act as an "one-time authentication key" for the transaction. This *nonce* is removed from the blockchain if the transaction uses the *nonce* (one-time use), or after a period of time (timeout). From BN's perspective, it is one of the effective ways to verify transactions. When the transaction reaches BN as a final step, BN will not refuse the execution as long as it is a completed form signed by TN. For this reason, the transaction packet can be copied and sent over and over even if it has already been sent and there is no such *nonce* value. Through the above process, it is possible to determine whether the packet is forged or not, and also the integrity of the transaction can be ensured. In addition, this process can be executed in parallel since multiple SNs with guaranteed integrity would work on concurrently.

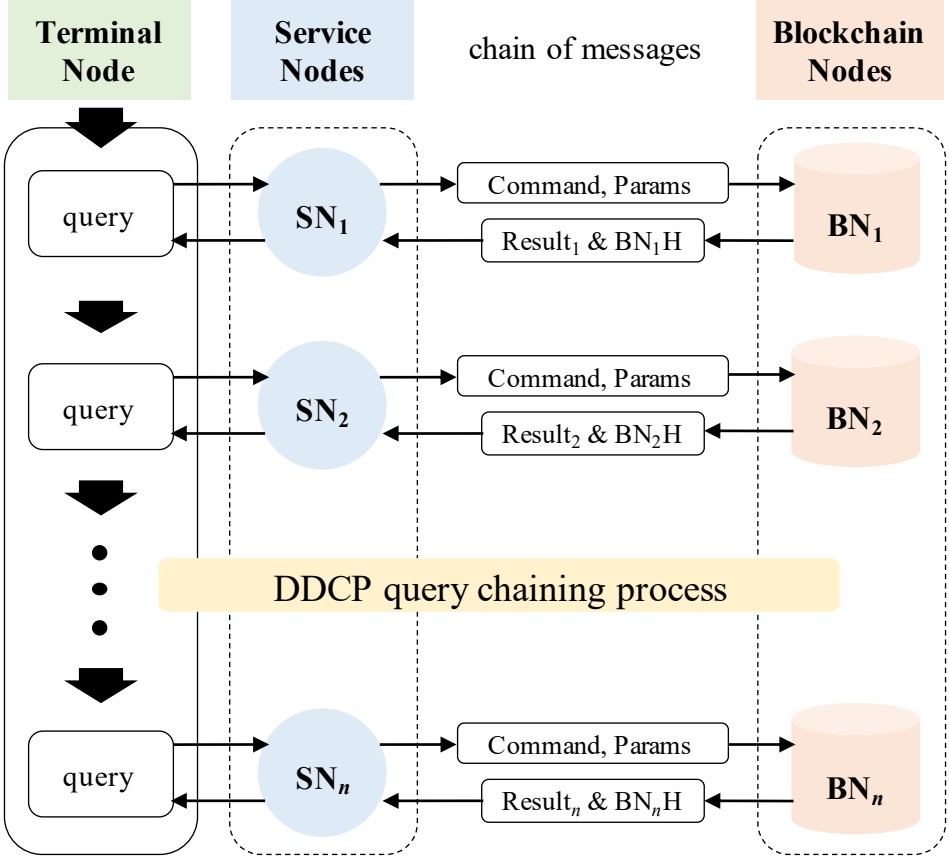

**Figure 2.** Dynamic Decentralized Certification Protocol (DDCP).

*4.3. DDNC (Dynamic Decentralized Network Consensus)*

To implement the service layer as an overlay network on top of a permissioned blockchain layer, the authenticity of all service nodes must be verified first. As anyone can participate in service layer, a verification process to ensure SNs' reliability and trustworthiness is essential. SN should act as intended by the application business logic, and every SN in a single domain acts like a clone of itself. This means that all SNs must always output a consistent value given some input. Based on this fact, a simple execution and comparison is carried out between SNs to determine whether they behave identically. This verification process is named DDNC (Dynamic Decentralized Network Consensus).

In DDNC, if a new node wishes to participate in the service layer, it will be verified by preexisting nodes (verifier nodes) as shown in Figure 3. Candidate node sends a request for the list of SNs to seed node, then asks for 'add node' permission to all SNs in the list. Each verifier node examines whether the candidate node behaves identically or not via the 'Service Node Verification Protocol' in Figure 4, and adds it to the host list if the candidate node behaves equally to a verified node. Otherwise, it will be purged. This 'peer-review' process runs not only when 'add node' is requested, but also periodically to make sure that all nodes continue to act as intended. Therefore, service layer can detect abnormal SNs and remove them from the network automatically and autonomously at runtime. With this verification process, SNs are assured and confirm the share of same domain and mutual trust. Furthermore, it gives service providers the advantage of being able to scale out their SNs flexibly while reducing unexpected risks.

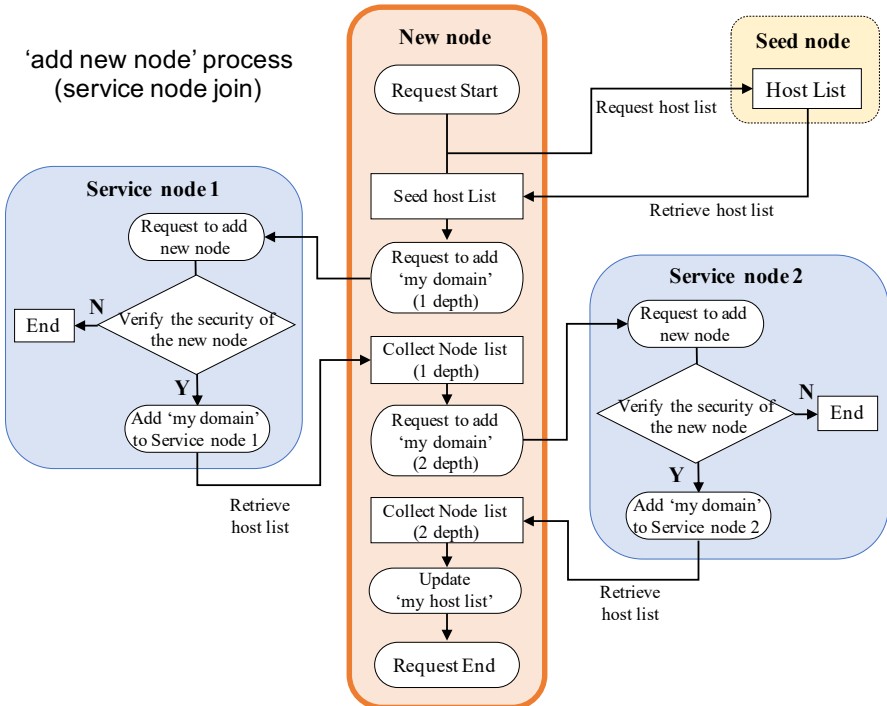

**Figure 3.** The process of 'adding new node'.

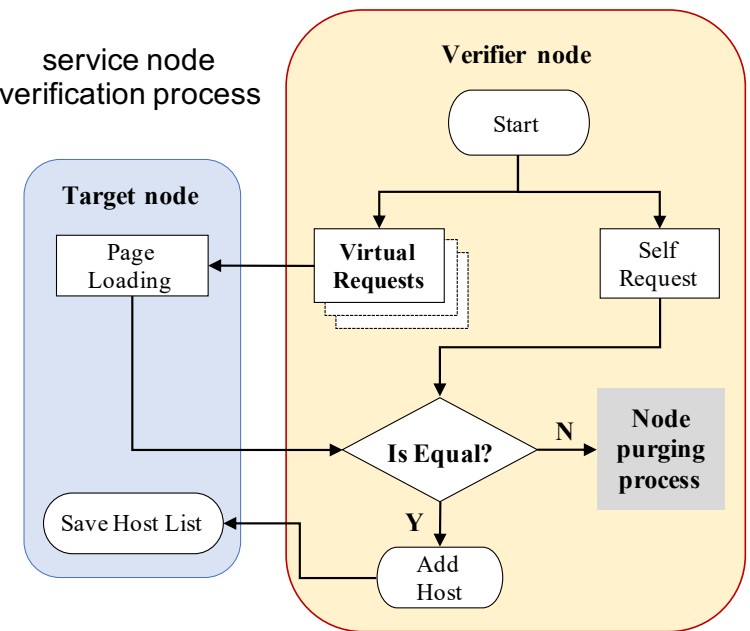

**Figure 4.** Service Node Verification Protocol.

### 4.4. SN-BN Communication Process

One final component of our layered architecture is the node agents between SN and blockchain network. Node agents are mediators between two networks and all accesses from SNs to the blockchain network pass through the node agents. There can be multiple node agents. If a node agent receives a request from an unauthorized SN, the node agent registers the SN into a blacklist and blocks additional requests for a certain period of time. In addition, node agents serve as a load balancer to prevent load concentration in one blockchain node and as a dispatcher of events from BNs.

Finally, using this 3-layer architecture, DDNB implements various functions to support application business logic. For example, TN can connect to a randomly selected SN (with function '*FindHost*'),

TN can get *nonce* from BN through SN (with function '*GetNonce*'), TN can get an account balance of specific address (with function '*GetBalance*'), and TN can send a coin amount (virtual currency) from an account to another one (with function '*SendCoin*').

- *FindHost* is the function that TN uses to connect to randomly selected SN through seed node. As the function does not reach BN, execution time is expected to be fast and stable. Changes in the number of SNs is what the function is most sensitive to.
- *GetNonce* requests a *nonce*, a hashed string sent from BN, to identify the transactions which are disassembled by several steps. Since any step in any transaction is processed on a randomly selected SN, BN publishes a tiny hashed string for discrimination.
- *GetBalance* is a function that returns the balance of an account. It internally executes *FindHost* twice, *GetNonce* once, and then finally a *GetBalance* query. This represents a single step transaction that queries the network several times.
- *SendCoin* is the most representative transaction which invokes changes in the blockchain. It consists of *InfoTrans* query which validates syntactic errors first, and then *CreateTrans* query which validates semantic errors and executes the function as a final step. Every steps of the process is shown in Figure 5 in detail.

API documentations of other functions will be available on our website at [40].

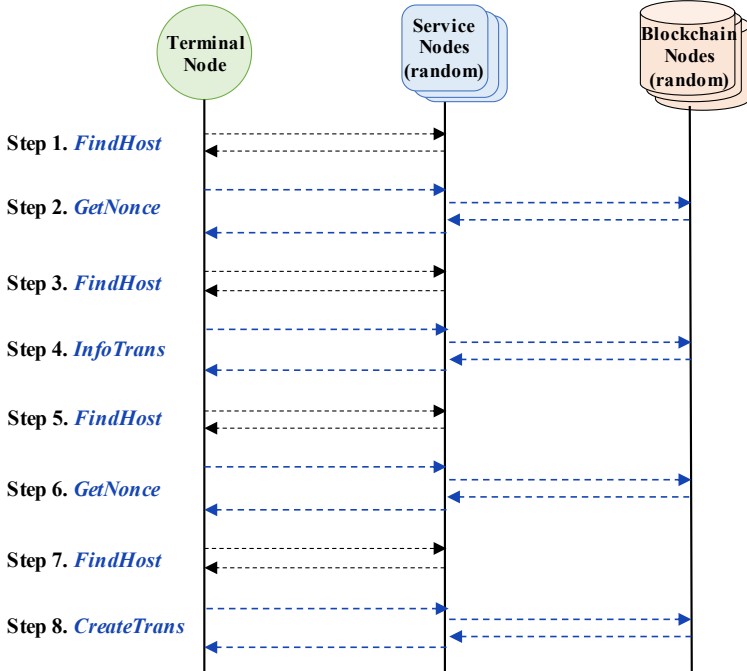

**Figure 5.** Detailed query steps of *SendCoin* function.

## 5. Performance Evaluation

This section presents the methodology and evaluation results of DDNB. Two properties of DDNB are investigated in this evaluation; performance and stability.

### 5.1. Experiment Setup

The environment on which the experiments are conducted are as follows. Terminal Nodes run on a desktop PC with Intel(R) Core(TM) i5-4670 CPU @ 3.40 GHz and 8 GB RAM, with Windows 10 Pro x64 multi-threading as the operating system. Service Nodes and Blockchain Nodes are established on four different enterprise level servers. Service layer is composed with Docker containers to run a large number of SNs. Blockchain layer is built as a permissioned blockchain based on Hyperledger.

On this setup Figure 6, four most frequently used functions (among a larger set) were chosen to evaluate DDNB as follows: *FindHost*, *GetNonce*, *GetBalance*, and *SendCoin*. In order to analyze the correlation between execution time and the number of steps a function has, each function was chosen according to the number of query steps they require.

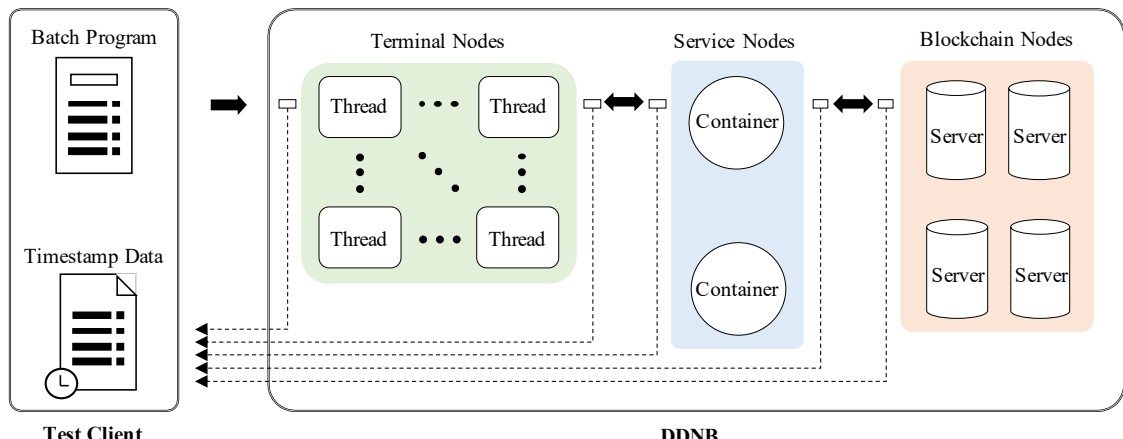

**Figure 6.** Test flow: Test client runs a batch program that executes a predefined transaction and records UNIX timestamps of every moment when packets are transmitted and received between nodes.

Each function is executed total of 1000 times by multiple TNs (end-users) on various topologies having different sets of TNs and SNs. All functions are executed asynchronously, that is, all requests are submitted without waiting for response from BN. The number of TNs are set to 5, 10, 20, 50, 100, 250, 500, and 1000 (emulating the number of end users), and SNs 1, 5, 10, and 20 (emulating service provider servers). The interactions between TN and SN are accomplished with HTTP requests on C++ application, and all queries between SN and BN are transmitted upon the RESTful APIs.

We assess the performance of DDNB in term of the *average execution time* and *transaction per second (TPS)*. For this purpose, following data are collected for each transaction.

- **Transaction deployment time** ($T_{deploy}$) is the time when a transaction is deployed to TN (beginning of execution).
- **Transaction completion time** ($T_{complete}$) is the time when a transaction completed the whole process.
- **Step (p) start time** ($T_{p\_start}$) and **Step (p) end time** ($T_{p\_end}$) are the start and end times of every step *(p)* in a transaction (when the request was responded from other node). Although detailed records have been made to determine the cause of outliers, no step-by-step analysis is conducted for the performance evaluation of the overall architecture.

Then for each topology, execution time is the total amount of time which the system took to execute and confirm all transactions in the data set ($T_{execute} = \sum_{i=1}^{1000} T_{deploy} - T_{complete}$). Since we are more interested in the average *execution time* for each TN, we divide it by the number of TNs performed in the experiment.

### 5.1.1. Average Execution Time

Figure 7 plots the execution time of 1000 transactions experimented on various topologies for the four functions. Commonly, the average execution time decrease as the number of TNs increases (thanks to increase in parallel, simultaneous execution of transactions), whereas the change with increasing SN is not striking. This is an interesting result as we were concerned that a large number of SNs might result in slower execution time due to validation overhead of DDNC. It turns out that, thanks to parallel execution, the overhead of DDNC does not scale proportionally with the number of SNs. The outliers for *GetNonce* and *GetBalance* are from packet drops between SN and BN due to

authentication failure. For example, if they wait for a *nonce* that has already expired or disappeared, the transaction will wait until timeout before it retransmits. This is an obvious problem observed in the experiments and worth a future work. *SendCoin*, on the other hand, shows similar aspects overall with no noticeable singularities, although it has the largest *average execution time* in every experiment due to its complexity (in terms of query steps required).

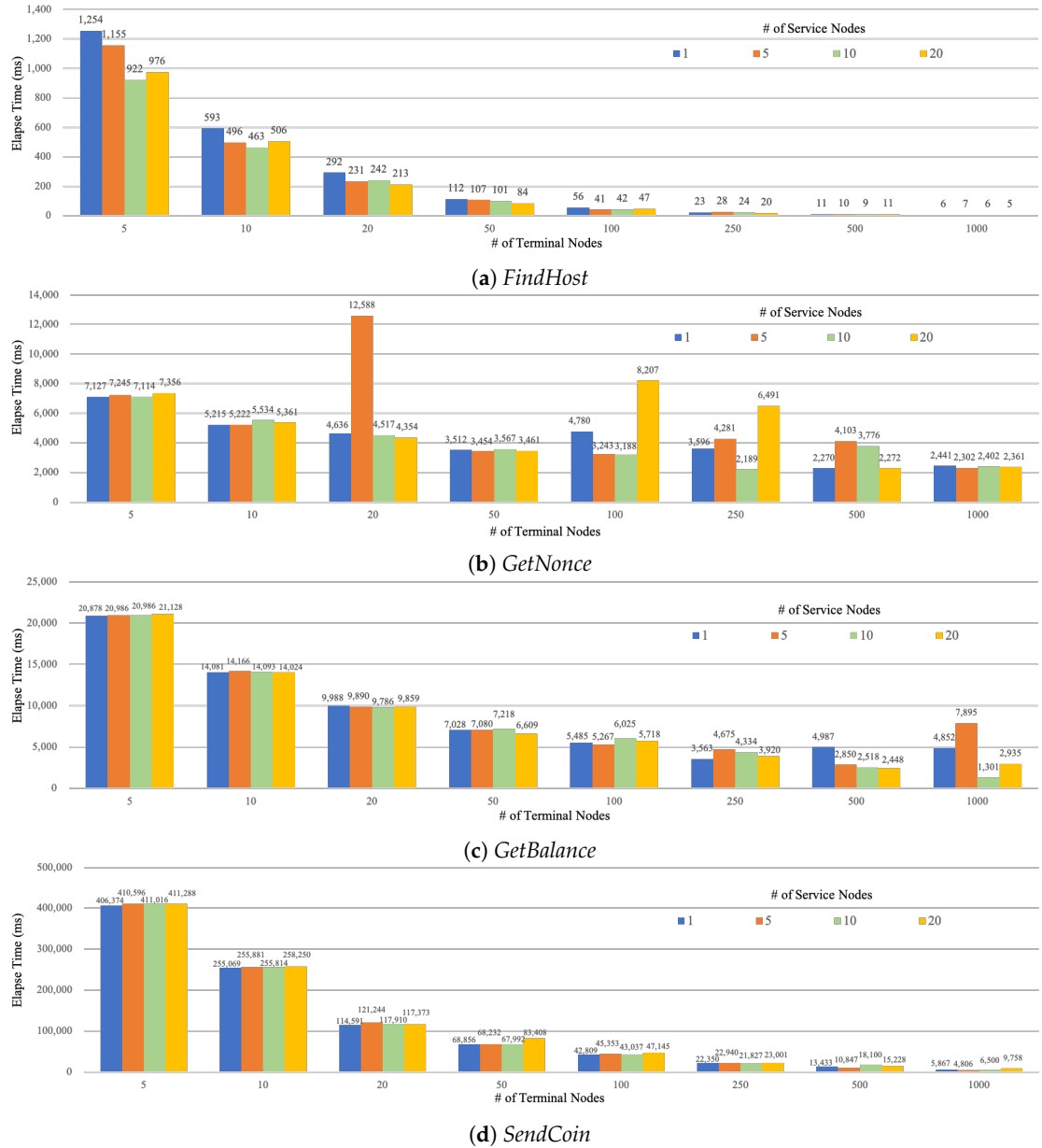

**Figure 7.** Average execution time over all topologies for four core functions: *FindHost*, *GetNonce*, *GetBalance*, and *SendCoin*. X-axis denote the number of Terminal Nodes (TNs), Y-axis is time in units of milliseconds, and the four colored bars represent the number of Service Nodes (SNs).

5.1.2. Transactions Per Second

*TPS* is the most commonly used indicator for network performance evaluation in blockchain architectures. Table 2 presents the *TPS* results for the *GetBalance* and *SendCoin* functions from the same experiment in Figure 7. It shows that in general, TPS is higher with larger number of TNs thanks to parallelism and randomization of SN selection, and the number of SNs do not have

direct correlation with TPS due to the same reasons as the execution time; that is, verification overhead from the increase in the number of SNs is cancelled out thanks to the improvement from parallelism.

**Table 2.** Transactions per second (TPS) of *GetBalance* and *SendCoin*.

| SN \ TN | 5 | 10 | 20 | 50 | 100 | 250 | 500 | 1000 |
|---|---|---|---|---|---|---|---|---|
| **GetBalance** | | | | | | | | |
| 1 | 47.9 | 71.0 | 100.1 | 142.3 | 182.3 | 280.7 | 200.5 | 206.1 |
| 5 | 47.7 | 70.6 | 101.1 | 141.3 | 189.9 | 213.9 | 350.9 | 126.7 |
| 10 | 47.7 | 71.0 | 102.2 | 138.5 | 166.0 | 230.7 | 397.2 | 768.8 |
| 20 | 47.3 | 71.3 | 101.4 | 151.3 | 174.9 | 255.1 | 408.5 | 340.7 |
| **SendCoin** | | | | | | | | |
| 1 | 2.5 | 3.9 | 8.7 | 14.5 | 23.4 | 44.7 | 74.4 | 170.4 |
| 5 | 2.4 | 3.9 | 8.2 | 14.7 | 22.0 | 43.6 | 92.2 | 208.1 |
| 10 | 2.4 | 3.9 | 8.5 | 14.7 | 23.2 | 45.8 | 55.2 | 153.9 |
| 20 | 2.4 | 3.9 | 8.5 | 12 | 21.2 | 43.5 | 65.7 | 102.5 |

The combination with the highest in the fixed number of TNs is highlighted in gray.

### 5.2. Comparison with other Platforms

To better understand the performance characteristics of DDNB, we compared it with other blockchain application platforms that employ the concept of smart contract. Nasir et al. measured the performance of two different versions of Hyperledger Fabric: v0.6 and v1.0 [41]. Between the two versions, there was a major re-architecture in the overall platform, including changes in the consensus model and the addition of support for channels. They deployed a simple money transfer application (chaincode) which has both the 'invoke' and 'query' functions. Pongnumkul et al. measured the performance of two permissioned blockchain implementations—Ethereum and Hyperledger Fabric v0.6 [42]. While the main Ethereum platform uses a public blockchain network, the software is open-source and allows developers to configure the network to work as a private network, where only granted nodes can participate in the network.

The two studies evaluated performance using various metrics, and we decided to focus on a common metric for all studies; TPS of executing 1000 transactions. The functions used in the comparison are the two representative functions in all compared schemes: *invoke* which changes the state of the blockchain, and *query*, for simple queries. As the performance result data for Hyperledger Fabric v0.6 differ in the two studies, we used median values for comparison.

Table 3 presents the TPS of the transactions executed on the respective platforms. Since the Blockchain Layer of DDNB is based on Hyperledger, the difference in performance between Hyperledger is not remarkable. However, DDNB's query function (*GetBalance*) outperforms Ethereum by 10.5 times, and 6.05 times for the invoke function. The benefits of separating the business logic in DDNB can be maintained while there is no performance penalty.

**Table 3.** Average TPS of various platforms for query and invoke functions.

| Platforms \ Functions | Query | Invoke |
|---|---|---|
| DDNB | 768.8 | 208.1 |
| Ethereum | 73.2 | 34.4 |
| HLF v0.6 | 296.3 | 203.0 |
| HLF v1.0 | 461.0 | 185.0 |

### 5.3. Stability Assessment

We now evaluate the stability of DDNC. Since DDNC's verification protocol is based on filtering out abnormalities among 'normal clones', the entire SN system can be subverted if the abnormal nodes occupy a 51% majority simultaneously. Therefore, the shorter the time it takes to execute the verification protocol, the more overhead it may have, but the more robust it is. In order to conduct experiments to evaluate the stability of DDNC, we measure how much time it took to restore after applying an intentional forging for arbitrary SNs in a single domain of the Service layer. Each experiment was conducted to obtain the time taken to blacklist all forged SNs with a DDNC set to 10-minute periods. Then we randomly and deliberately forged SNs among normal SNs.

Table 4 shows the average time taken for the experiments which were performed 10 times respectively. Intuitively, the more nodes are forged, the longer it takes to cleanse the entire system. Not only it has more forgeries to find, but it takes more time for each node because larger ratio of forgery makes more difficult to determine whether it is normal or forged.

**Table 4.** Average time (in seconds) to blacklist forged nodes.

| SN \ Forged | 1 | 3 | 9 |
|---|---|---|---|
| 1 | - | - | - |
| 5 | 262 | - | - |
| 10 | 133 | 458 | - |
| 20 | 129 | 387 | 777 |

As can be seen from Figure 8, the case of (a) *9 forgeries in 20 SNs* is mostly blacklisted in the first period except the last or second node. On the other hand, in case of (b) *3 forgeries in 20 SNs* which forged less number of SNs, DDNC blacklists all forged nodes within first period in every experiments. This result is consistent with when comparing (b) and (c). Also in (c) *3 forgeries in 10 SNs*, several experiments were observed that took more than one period to find the last forgery.

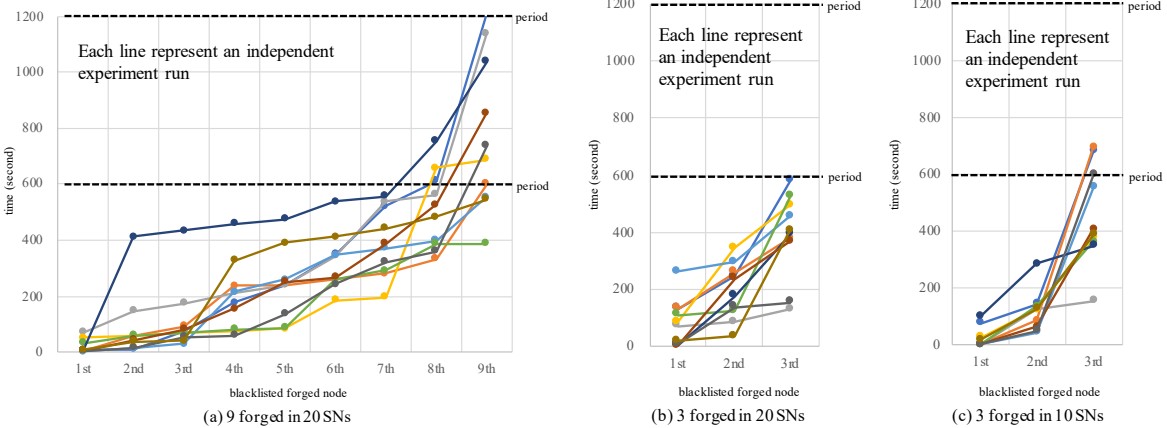

**Figure 8.** Completion time of blacklisting (mutual verification process) with varying number of forged nodes—results from 10 independent experiment runs.

## 6. Application Case Study

We now present an application case study using DDNB, implementations of sample application services that benefit from the use of DDNB architecture.

*Nodehome* is a development environment for blockchain-based applications and also an operating platform for running developed services on blockchain. Even without a complex understanding of blockchain, developers are able to implement their own services according to the API rules

(Terminal Layer ↔ Service Layer) provided by the *Nodehome* platform [40]. Developers need not worry about the terminals on which they run their services, nor how to prepare the hardware and network infrastructure for their blockchain. If you can provide hardware that can run pre-developed services, you can share the profits generated by services on *Nodehome* platform. *Nodehome* platform is equipped with various blockchain based means to ensure trust, such as proof of assets, preservation of goods, movement of goods, registration of property, evidence of actions, and protection of information. Many blockchain-based services are currently being implemented and executed on the *Nodehome* platform, and the information and assets are exchangeable between different services. *Nodehome* platform is serviced currently in the form of a testnet, and web pages (https://nodehome.io/) provide block explorers that provides detailed information about blocks, addresses, and transactions. In addition, a simple guide for architects, developers, and users is provided as well.

As shown in Figure 9, a few client applications are also available. *NH Token Wallet* is a wallet service that manages tokens issued from various applications in the DDNB's Service Layer (of course, some applications may not provide token-related services intentionally). *fMusic*-music streaming service is one of the practical applications built on DDNB. Since it is provided as a distributed service contrasting to existing services, it is able to connect singer-to-listener without sound production or distribution company. Not only this simple difference, but in intellectual property or derived user data perspective, it can also seek several new attempts that have not been possible before.

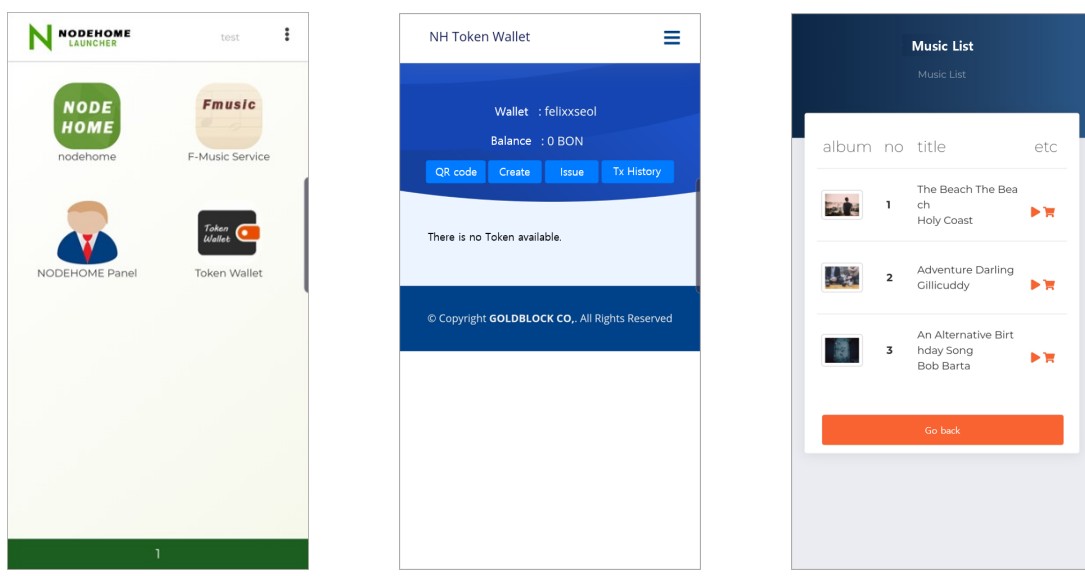

**Figure 9.** Screenshots of *Nodehome* mobile applications. From the left are the launcher main, wallet, and *fMusic*-music streaming services.

## 7. Conclusions

This work presented DDNB, a blockchain-based application service platform that allows application developers to take advantage of the integrity and reliability of blockchain technology while tackling the disadvantages of existing blockchain systems. DDNB consists of three layers: Application clients run in the public Terminal layer on top of the Service network, and accesses the permissioned Blockchain network only through the service network. DDNB enhances the reliability of the system through our proposed *self-regulating mutual node verification* and distributed *query-chain* process while reducing the consensus delay of blockchain with the aid of the unique service layer that separates the applications from the permissioned blockchain layer. Moreover, developers are not required to have deep understandings of the blockchain technology to develop blockchain based services on the Terminal layer, and end-users do not need to maintain blockchain nodes by themselves for using the services. With all of these benefits, the performance comparisons with other blockchain platforms showed that DDNB has no performance penalty, and the robustness of DDNC against

random malicious attack was also evaluated. We also presented real-world application case studies that are currently being developed and operated. We anticipate various invigorating blockchain based services to emerge via DDNB.

**Author Contributions:** Conceptualization, H.C.; Formal analysis, Y.S.; Project administration, J.P.; Resources, Y.S. and M.J.; Software, J.Y.; Supervision, J.P.; Validation, Y.S.; Visualization, Y.S.; Writing—original draft, Y.S., J.A. and S.P.; Writing—review & editing, Y.S. and J.P. All authors have read and agreed to the published version of the manuscript.

**Funding:** This research was supported by the MIST (Ministry of Science and ICT), Korea, under the National Program for Excellence in SW(20170001000041001) supervised by the IITP (Institute of Information & communications Technology Planning & Evaluation), and also by the Chung-Ang University Graduate Research Scholarship in 2019.

**Conflicts of Interest:** The authors declare no conflicts of interest. The funders had no role in the design of the study; in the collection, analyses, or interpretation of data; in the writing of the manuscript; or in the decision to publish the results.

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
