# Peer review of "DDNB—Doubly Decentralized Network Blockchain Architecture for Application Servicesâ€"

_applsci, doi:10.3390/app10155212_

Round 1
Reviewer 1 Report
The main concepts were already presented in the previous publication. This paper adds the implementation and test of the concept and most of the effort should be concentrated in this new contribution. So, readers would benefit from having a more detailed discussion on the results presented. For example: it is not clear why the execution time decreases with the increase of the number of TNs. It seems that the number of SNs does not affect the execution time. Why is this? What is relevant in table 2? How can the results be explained?
Author Response
The main concepts were already presented in the previous publication. This paper adds the implementation and test of the concept and most of the effort should be concentrated in this new contribution. So, readers would benefit from having a more detailed discussion on the results presented. For example: it is not clear why the execution time decreases with the increase of the number of TNs. It seems that the number of SNs does not affect the execution time. Why is this?
[Our response]: The purpose of the experiment was to measure and compare the execution time for completing 1000 transactions on each TN-SN topology. With the increase in the number of TNs, more number of trans- actions can be processed simultaneously in parallel on larger number of TNs, and this is the reason why the execution time decreases with the increase of the number of TNs. On the other hand, when the number of SNs increase, DDNB has two opposite effects. Increase in the number of SNs increases the number of validation steps and its corresponding overhead, and thus is expected to increase the execution time. On the contrary, it also increases the parallelism of the SN processing, thus reducing the execution time. These two effects together cancels each other and results in similar overall execution time. We have revised the text to explain this, and we hope this answers the question raised by the reviewer.
What is relevant in table 2? How can the results be explained?
[Our response]: Table 2 is the result from the same experiment as Figure. 7, and they have similar implications; performance improves with the increase in number of TNs, and the number of SNs do not have significant impact. The difference is that, Figure.7 was in terms of execution time, and Table 2 is in the metric of TPS (transactions per second). The explanation for such result is identical to the reviewers previous question. We hope this answers the question asked by the reviewer.

Reviewer 2 Report
Interesting work that can be used in practical applications. In my opinion, it deserves to be published in its current form
Author Response
Interesting work that can be used in practical applications. In my opinion, it deserves to be published in its current form.
[Our response]: We thank the reviewer greatly for recommending publication of our work. We really appreciate it.

Reviewer 3 Report
The authors present the performance evaluation of their proposed architecture regarding a blockchain based platform for application services. The overall architecture has been described on a previous paper of theirs and now they provide the performance results proving that their approach is suitable for application deployment based on blockchain technology. The manuscript is well-written and easy for the reader to understand it. However, there are some minor issues that should be addressed before the publication.
1) The manuscript provides significant new content in contrast with their previous work presented in “Seol, Y.; Ahn, J.; Park, S.; Ji, M.; Chae, H.; Yi, J.; Kim, Y.; Paek, J. Query-chain: Fast and Flexible Blockchain-based Platform for Diverse Application Services. International Conference on Information and Communication Technology Convergence (ICTC), 2019” but there are some text and figures that are identical between these two papers. So, some effort is needed in order to rephrase these parts.
2) In the chart axes of Figure 7, the measurement units are missing.
3) The authors should consider moving section 6 (Related work) after section 2 (Problem motivation)
4) At the end of section 3, the authors state that the API documentation will be available on their website, but no link is provided.
Author Response
The authors present the performance evaluation of their proposed architecture regarding a blockchain based platform for application services. The overall architecture has been described on a previous paper of theirs and now they provide the performance results proving that their approach is suitable for application deployment based on blockchain technology. The manuscript is well-written and easy for the reader to understand it. However, there are some minor issues that should be addressed before the publication.
1) The manuscript provides significant new content in contrast with their previous work presented in Seol, Y.; Ahn, J.; Park, S.; Ji, M.; Chae, H.; Yi, J.; Kim, Y.; Paek, J. Query-chain: Fast and Flexible Blockchain-based Platform for Diverse Application Services. International Conference on Information and Communication Technology Convergence (ICTC), 2019 but there are some text and figures that are identical between these two papers. So, some effort is needed in order to rephrase these parts.
[Our response]: We have modified both the text and figures so that they are not identical to our prior publication in ICTC 2019. Before the modification, the similarity of the initial version of the submitted manuscript to the conference version was 16% in iThenticate, which is well below the widely accepted level for conference-to- journal extensions. After the modification, we believe it is significantly lower. We hope this resolves the concern raised by the reviewer.
2) In the chart axes of Figure 7, the measurement units are missing.
[Our response]: We apologize for our mistake. We have added the missing measurement units in the chart axes of Figure 7.
3) The authors should consider moving section 6 (Related work) after section 2 (Problem motivation)
[Our response]: We thank the reviewer for the suggestion. We have moved the related work section to section 3 after the problem motivation section, as per recommended by the reviewer.
4) At the end of section 3, the authors state that the API documentation will be available on their website, but no link is provided.
[Our response]: We thank the reviewer great for pointing out our mistake. We have added the link to the API documentation in the revised version of the manuscript.

Reviewer 4 Report
- Authors are required to read the literature to see what's new out there. hence, the references provided in the paper are outdated (no references from 2020 and only 1 reference from 2019). They need to be updated.
- Rather than public/private block chain systems, they are categorized as permission-ed/permission-less.
- In this kind of submissions (improvement on a conference paper); authors are expected to submit a report that explains/enlists what are the improvements and enhancements in the newer version. This is missing!
Author Response
Authors are required to read the literature to see what’s new out there. hence, the references provided in the paper are outdated (no references from 2020 and only 1 reference from 2019). They need to be updated.
[Our response]: We thank the reviewer for pointing out our mistake, and we apologize for not updating our references. In the revised manuscript, we have read, cited, and added several new references that are from 2019 and 2020. We have updated the related work, introduction, and the motivation section accordingly. We hope that our revision addresses the concern raised by the reviewer.
Rather than public/private block chain systems, they are categorized as permission-ed/permission-less.
[Our response]: We have modified the terms public/private to permissioned/permissionless as suggested by the reviewer. We thank the reviewer for the suggestion.
In this kind of submissions (improvement on a conference paper); authors are expected to submit a report that ex- plains/enlists what are the improvements and enhancements in the newer version. This is missing!
[Our response]: We apologize for not being clear on this matter. In the initial version of the manuscript, we wrote the following sentence on the first page; “An earlier preliminary version of this idea was presented at the International Conference on ICT Convergence (ICTC) 2019 [1]. This manuscript is a complete re-write, and adds new content including implementation, evaluation, and case-study of the system.” The conference paper only had the initial idea. This journal version significantly extends the conference version where the key differences are: (1) we have implemented our idea into a real system on real hardware, (2) we have evaluated the performance of our system on several topologies with varying number of SNs and TNs for several core functions of our system, (3) we have conducted a case-study of our system, and (4) we have extended the intro, problem/motivation, and related work to provide better understanding of the subject. We hope this addresses the question asked by the reviewer.

Round 2
Reviewer 4 Report
- There is no scientific discussion provided to support the ideas provided in Table-1! Please elaborate.
- Figure-1 cropped from the top! Double check!
- It is better to use permissioned/permissionless terminology only for Blockchain layer. It is quite irrelevant for terminal layer.
- Use some other wording for this: "semi-permissioned service layer"
- p11L346: "Table 4 plots the average time taken", how can a table plot?
- In Figure-8 it is not clear what the plots from A to J represent. It would be better to extend discussion for this figure.
Author Response
Reviewer #4 - Comments and Suggestions for Authors:
There is no scientific discussion provided to support the ideas provided in Table-1! Please elaborate.
[Our response]: We thank the reviewer for the constructive comment. We have added extra explanation at the end of section II regarding our view and opinion on the properties of the four different application service architectures.
Figure-1 cropped from the top! Double check!
[Our response]: Dear reviewer, I’m not sure whether I understood your comment correctly, but I do not see any problem with Figure 1 being cropped. It may be possible that the problem is in the pdf generation of the MDPI manuscript submission system. We have double checked the figure, and it looks okay to us. We will attach our pdf version of the manuscript to this letter.
It is better to use permissioned/permissionless terminology only for Blockchain layer. It is quite irrelevant for terminal layer.
[Our response]: We thank the reviewer for the suggestion. We have modified the text so that the terms ‘permis- sioned/permissionless’ is used only for the blockchain layer and not for the terminal layer.
Use some other wording for this: ”semi-permissioned service layer”
[Our response]: We thank the reviewer for the comment. We have removed that wording, and explained it in words saying that the service layer is an interface that separates the permissioned blockchain layer from the public terminal layer.
p11L346: ”Table 4 plots the average time taken”, how can a table plot?
[Our response]: We apologize for our typo. We have corrected the word ‘plot’ to ‘show’. We thank the reviewer for pointing out our mistake.
In Figure-8 it is not clear what the plots from A to J represent. It would be better to extend discussion for this figure.
[Our response]: We apologize for our mistake. It should have been ‘independent experiment runs 1, 2, 3, ...’ and so on. We have corrected it in the revised version of the manuscript.
